# An equation-of-state-meter of quantum chromodynamics transition from deep learning

Long-Gang Pang [1,2,3], Kai Zhou [1,4], Nan Su[1], Hannah Petersen [1,4,5], Horst Stöcker[1,4,5] & Xin-Nian Wang [3,6]

A primordial state of matter consisting of free quarks and gluons that existed in the early universe a few microseconds after the Big Bang is also expected to form in high-energy heavy-ion collisions. Determining the equation of state (EoS) of such a primordial matter is the ultimate goal of high-energy heavy-ion experiments. Here we use supervised learning with a deep convolutional neural network to identify the EoS employed in the relativistic hydrodynamic simulations of heavy ion collisions. High-level correlations of particle spectra in transverse momentum and azimuthal angle learned by the network act as an effective EoS-meter in deciphering the nature of the phase transition in quantum chromodynamics. Such EoS-meter is model-independent and insensitive to other simulation inputs including the initial conditions for hydrodynamic simulations.

[1] Frankfurt Institute for Advanced Studies, 60438 Frankfurt am Main, Germany. [2] Department of Physics, University of California, Berkeley, CA 94720, USA. [3] Nuclear Science Division, Lawrence Berkeley National Laboratory, Berkeley, CA 94720, USA. [4] Institut für Theoretische Physik, Goethe Universität, 60438 Frankfurt am Main, Germany. [5] GSI Helmholtzzentrum für Schwerionenforschung, 64291 Darmstadt, Germany. [6] Key Laboratory of Quark and Lepton Physics (MOE) and Institute of Particle Physics, Central China Normal University, Wuhan 430079, China. Correspondence and requests for materials should be addressed to L.-G.P. (email: lgpang.1984@berkeley.edu) or to K.Z. (email: zhou@fias.uni-frankfurt.de) or to N.S. (email: nansu@fias.uni-frankfurt.de)

Deep learning (DL) is a branch of machine learning that learns multiple levels of representations from data[1,2]. DL has been successfully applied in pattern recognition and classification tasks, such as image recognition and language processing. Recently, the application of DL to physics research is rapidly growing, such as in particle physics[3–7], nuclear physics[8], and condensed matter physics[9–14]. DL is shown to be very powerful in extracting pertinent features especially for complex non-linear systems with high-order correlations that conventional techniques are unable to tackle. This suggests that it could be utilized to unveil hidden information from the highly implicit data of heavy-ion experiments.

Strong interaction in nuclear matter is governed by the theory of quantum chromodynamics (QCD). It predicts a transition from the normal nuclear matter, in which the more fundamental constituents, quarks and gluons, are confined within the domains of nucleons, to a new form of matter with freely roaming quarks and gluons as one increases the temperature or density. The QCD transition is conjectured to be a crossover at small density (and moderately high temperature), and first order at moderate density (and lower temperature), with a critical point separating the two, see Fig. 1 for a schematic QCD phase diagram and[15–17] for some reviews. One primary goal of ultra-relativistic heavy-ion collisions is to study the QCD transition.

Though it is believed that strongly coupled QCD matter can be formed in heavy-ion collisions at the Relativistic Heavy Ion Collider (RHIC, Brookhaven National Laboratory, USA)[18], Large Hadron Collider (LHC, European Organization for Nuclear Research, Switzerland)[19], and at the forthcoming Facility for Anti-proton and Ion Research (FAIR, GSI Helmholtz Centre for Heavy Ion Research, Germany)[20,21], a direct access to the bulk properties of the matter such as the equation of state (EoS) and transport coefficients is impossible due to the highly dynamical nature of the collisions. In heavy-ion collisions where two high-energy nuclei collide along the longitudinal ($z$) direction, what experiments measure directly are the final-state particle distributions in longitudinal momentum (rapidity), transverse momentum $p_T$ and azimuthal angle $\phi$.

Current efforts to extract physical properties of the QCD matter from experimental data are through direct comparisons with model calculations of event-averaged and predefined observables, such as anisotropic flow[22] or global fitting of a set of observables with Bayesian method[23,24]. However, event-by-event

raw data on $\rho(p_T, \phi)$ at different rapidities provide much more information that contains hidden correlations. These hidden correlations can be sensitive to physical properties of the system but independent of other model parameters.

The aim of the present exploratory study is a first step in directly connecting QCD bulk properties and raw data of heavy-ion collisions using state-of-the-art deep-learning techniques. We use the relativistic hydrodynamic model which has been very successful in simulating heavy-ion collisions and connecting experiments with theory[25–29]. We find unique encoders of bulk properties (here we focus on the EoS) inside $\rho(p_T, \phi)$ in terms of high-level representations using deep-learning techniques, which are not captured by conventional observables. This is achieved by constructing a convolutional neural network (CNN) and training it with labeled $\rho(p_T, \phi)$ of charged pions generated from the relativistic hydrodynamic program CLVisc[30,31] with two different EoSs as input: crossover[32] and first order[33]. The CNN is then trained with supervision in identifying different EoSs. The performance is surprisingly robust against other simulation parameters such as the initial conditions, equilibrium time $\tau_0$, transport coefficients and freeze out temperature. The supervised learning with deep CNN identifies the hydrodynamic response which is much more tolerant to uncertainties in the initial conditions. $\rho(p_T, \phi)$ as generated by independent simulations (CLVisc with different setup parameters and another hydrodynamic package iEBE-VISHNU[34] which implements a different numerical solver for partial differential equations) are used for testing—on average a larger than 95% testing accuracy is obtained. It has been recently pointed out that model-dependent features (features in the training data that depends on the simulation model and parameters) may generate large uncertainties in the network performance[6]. The network we develop below is, however, not sensitive to these model-dependent features.

## Results

**Training and testing data sets**. The evolution of strongly coupled QCD matter can be well described by second-order dissipative hydrodynamics governed by $\partial_\mu T^{\mu\nu} = 0$, with $T^{\mu\nu}$ the energy–momentum tensor containing viscous corrections governed by the Israel–Stewart equations[25,26]. In order to close the hydrodynamic equations, one must supply the EoS of the medium as one crucial input. The nature of the QCD transition in the EoS strongly affects the hydrodynamic evolution[35], since different transitions are associated with different pressure gradients which consequently induce different expansion rates, see the small chart in Fig. 1. Final $\rho(p_T, \phi)$ are obtained from the Cooper–Frye formula for particle $i$ at mid-rapidity

$$\rho(p_T, \phi) \equiv \frac{dN_i}{dY p_T dp_T d\phi} = g_i \int_\sigma p^\mu d\sigma_\mu f_i, \tag{1}$$

Here $N_i$ is the particle number density, $Y$ is the rapidity, $g_i$ is the degeneracy, $d\sigma_\mu$ is the freeze-out hypersurface element, $f_i$ is the

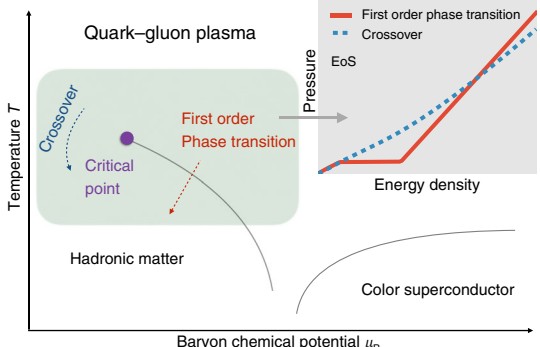

**Fig. 1** The conjectured phase diagram in quantum chromodynamics. In the region with high temperature and small baryon chemical potential, the phase transition between hadronic matter and quark–gluon plasma is a cross over according to lattice QCD calculations (blue dashed line in the small insert). In the region with low temperature and moderately high baryon chemical potential, the phase transition is first order (red line in the small insert). At low temperature and high baryon chemical potential, there might exist other phases, such as color superconductor

| Training data set | $\eta/s = 0$ | | $\eta/s = 0.08$ | |
|---|---|---|---|---|
| | **EOSL** | **EOSQ** | **EOSL** | **EOSQ** |
| Au–Au $\sqrt{s_{NN}} = 200$ GeV | 7435 | 5328 | 500 | 500 |
| Pb–Pb $\sqrt{s_{NN}} = 2.76$ TeV | 4967 | 2828 | 500 | 500 |

**Table 1 The training data set**

Numbers of $\rho(p_T, \phi)$ generated by the CLVisc hydrodynamic package with the AMPT initial conditions in the centrality range 0–60%. $\eta/s$ is ratio of shear viscosity to entropy density. $\tau_0 = 0.4$ fm for the Au–Au collisions and $\tau_0 = 0.2$ fm for the Pb–Pb collisions. The freeze-out temperature is set to be 137 MeV

**Table 2 The testing data set**

**Testing data set group 1: iEBE-VISHNU + MC-Glauber**

| Centrality: 10–60% | $\eta/s \in [0, 0.05]$ | | $\eta/s \in (0.05, 0.10]$ | | $\eta/s \in (0.10, 0.16]$ | |
|---|---|---|---|---|---|---|
| | EOSL | EOSQ | EOSL | EOSQ | EOSL | EOSQ |
| Au–Au $\sqrt{s_{NN}} = 200$ GeV | 650 | 850 | 900 | 750 | 200 | 950 |
| Pb–Pb $\sqrt{s_{NN}} = 2.76$ TeV | 500 | 650 | 600 | 644 | 499 | 150 |
| Testing data set group 2: CLVisc + IP-Glasma | | | | | | |
| Au–Au $\sqrt{s_{NN}} = 200$ GeV, $b \lesssim 8$ fm | EOSL | | | EOSQ | | |
| $\eta/s = 0$ | 4164 | | | 4752 | | |
| $\eta/s = 0.08$ | 1173 | | | 864 | | |

Numbers of $\rho(p_T, \phi)$ generated by the CLVisc and iEBE-VISHNU hydrodynamic packages with different initial conditions. $\eta/s$ is ratio of shear viscosity and entropy density. $b$ is the impact parameter. $\tau_0 = 0.6$fm for all the collisions. In iEBE-VISHNU simulations, the freeze-out temperature is varied in the range [115, 142]MeV. In CLVisc simulations, the freeze-out temperature is set to be 137 MeV

thermal distribution. In the following, we employ the lattice-EoS parametrization[32] (dubbed as EOSL) for the crossover transition and Maxwell construction[33] (dubbed as EOSQ) for the first-order phase transition.

The training data set of $\rho(p_T, \phi)$ (labeled with EOSL or EOSQ) is generated by event-by-event hydrodynamic package CLVisc[30,31] with fluctuating AMPT initial conditions[36]. The simulation generated about 22,000 $\rho(p_T, \phi)$ for different types of collisions. Then the size of the training data set is doubled by label-preserving left-right flipping along the $\phi$ direction. In Table 1 we list the details of the training data set.

The testing data set contains two groups of samples. In the first group, we generate 7343 $\rho(p_T, \phi)$ events using the second-order event-by-event hydrodynamic package iEBE-VISHNU[34] with MC-Glauber initial condition. In the second group, we generate 10953 $\rho(p_T, \phi)$ events using the CLVisc package with the IP-Glasma-like initial condition[24,37]. The testing data sets are constructed to explore very different regions of parameters as compared to training data set. The details are listed in Table 2. Note that all the training and testing $\rho(p_T, \phi)$ are preprocessed by $\rho' = \rho/\rho_{max} - 0.5$ to normalize the input data.

**The existence of physical encoders and neural-network decoder.** After training and validating the network, it is tested on the testing data set of $\rho(p_T, \phi)$ events (see Sec. 4 for the details of our neural-network model). As shown in Table 3, high prediction accuracies—on average larger than 95% with small model uncertainties given by a 10-fold cross validation tests—are achieved for these three groups of testing data sets, which indicates that our method is highly independent of initial conditions. The network is robust against shear viscosity and $\tau_0$ due to the inclusion of events with different $\eta/s$ and $\tau_0$ in the training. In testing stage the neural network identifies the type of the QCD transition solely from the spectra of each single event. Furthermore, in the training only one freeze-out temperature is used, while the network is tolerant to a wide range of freeze-out temperatures during the testing. For simplicity, the exploratory study has not included pions from resonance decays (the hadronic transport module UrQMD is switched off in iEBE-VISHNU to exclude contributions from resonance decays in testing data).

For complex and dynamically evolving systems, the final states may not contain enough information to retrieve the physical properties of initial and intermediate states due to entropy production (information loss) during the evolution. The mean prediction accuracy decreases from 97.1% (for $\eta/s = 0.0$) to 96.6% (for $\eta/s = 0.08$) and 87% (for $\eta/s = 0.16$) in the 10-fold cross validation for testing GROUP 1. Besides, the construction of conventional observables may introduce further information loss due to projection of raw data to lower dimensions, as well as

**Table 3 Testing accuracies**

| Testing data | Group 0 | Group 1 | Group 2 |
|---|---|---|---|
| Number of events | 4000 | 7343 | 10,953 |
| Accuracy | 99.88 ± 0.04% | 93.46 ± 1.35% | 93.91 ± 3.92% |

The mean prediction accuracies and the standard deviations given by ten trained models in cross validation method, from three groups of testing data sets, (GROUP 0) CLVisc with AMPT initial condition, (GROUP 1) iEBE-VISHNU and (GROUP 2) CLVisc with the IP-Glasma-like initial condition

information interference due to its sensitivity to multiple factors. These make it yet unclear how to reliably extract physical properties from raw data. Our study firmly demonstrates how to detect the existence of physical encoders in final states with deep CNN decoders, and sets the stage for further applications, such as identifying all relevant physical properties of the systems.

**Observation from the neural-network decoder.** In order to get physical insights from the neural-network model, it is instructive to visualize the complex dependences learned by the network. For this purpose, we employ the recently developed Prediction Difference Analysis method[38,39]. This method uses the observation that replacing one feature in the input image can induce a sizable prediction difference if that feature is important for classification decision. The prediction differences can be visualized as the importance maps of all the input features for the classification network.

Shown in Fig. 2 are importance maps which illustrate the $(p_T, \phi)$ dependence of the mean prediction difference averaged over 800 events for different model setups (initial conditions, PDE solver and model parameters), EoSs and values of the shear viscosity. For a given event, the mean prediction difference in each $(p_T, \phi)$ bin is computed against ten random reference events from the same data set. Comparing different columns in the same row in Fig. 2, we can see that importance maps vary slightly for different values of viscosity and model setups (Group 1: IEBE-VISHNU + MC-Glauber, Group 2: CLVics + IP-Glasma) for the same EoS. However, importance maps for EOSL in general have a distinctly narrower width in the $p_T$ range than that for EOSQ, independently of the model setup and the value of viscosity[40]. This might be the important region of hidden features the network recognizes in classifying the EoS under each event.

**Discussion**

Besides the deep CNN method employed in the present paper, there are also some other machine learning classifiers. In Supplementary Note 2 we attached the results from several traditional

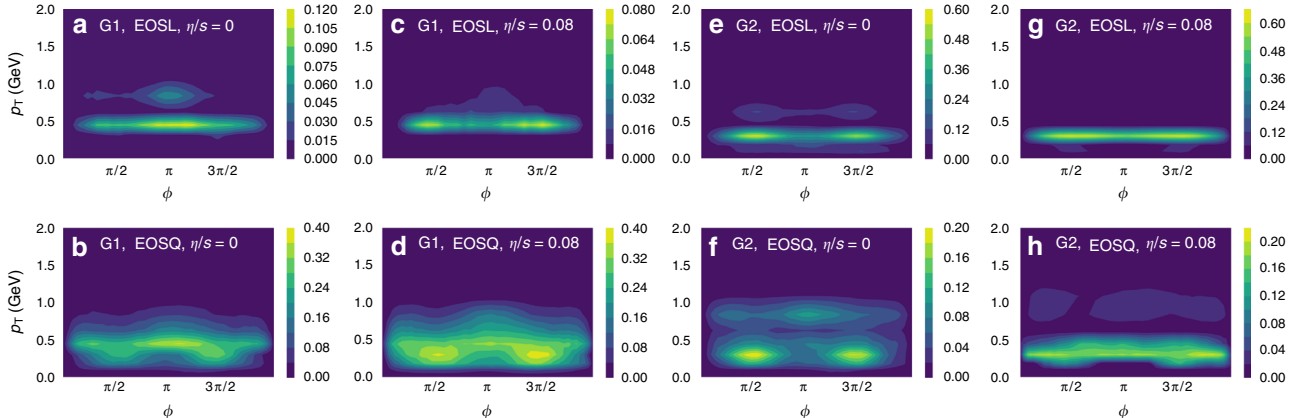

**Fig. 2** Importance maps of the particle momentum distribution. The values in the scale bar represent the relative importance of each bin for classification computed using the Prediction Difference Analysis method by averaging over about 800 events for each category. EOSL in the first row represents the equation of state with a smooth crossover, EOSQ in the second row represents a first-order phase transition equation of state. The G1 in **a**–**d** represents the testing data set Group 1 from iEBE-VISHNU model while G2 in **e**–**h** represents the testing data set from CLVisc + IP-Glasma model. The $\eta/s = 0$ in **a**, **b**, **e**, **f** represents ideal hydrodynamics while $\eta/s = 0.08$ in **c**, **d**, **g**, **h** represents viscous hydrodynamics with shear viscosity over entropy density ratio 0.08

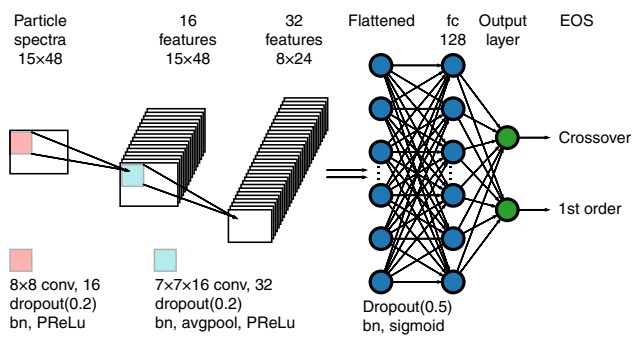

**Fig. 3** The convolution neural network architecture. The architecture is designed to identify the quantum chromodynamics transition by using particle spectra with 15 transverse momentum $p_T$ bins and 48 azimuthal angle $\phi$ bins

machine learning methods, such as support vector machine classifier (SVC), decision trees, random forests and gradient boosting trees. The best classifier (linear SVC) that generalizes well on two testing data sets achieves on average ~80% prediction accuracy. The important features from different classifiers differ from each other, however, those with good generalization capability have similar importance regions as given by the deep CNN. The deep CNN with on average ~95% prediction accuracy works much better to answer the core questions—is there a traceable encoder of the dynamical information from phase structure (EoS) that survives the evolution and exists in the final snapshot? If yes, then how to exclusively and effectively decode these information from the highly complex final output? These questions are crucial but unclear for decades in high-energy heavy-ion physics (and also in physical cosmology) due to the complexity and highly-dynamical characteristics in the collision evolution. The deep CNN demonstrates the revolution that big data analysis and machine learning might bring to the high energy physics and astrophysics.

The present method yields a perspective on identifying the nature of the QCD transition in heavy-ion collisions. With the help of deep CNNs and its well generalization performance, we firmly demonstrate that discriminative and traceable projections —encoders—from the QCD transition onto the final-state $\rho(p_T, \phi)$ do exist in the complex and highly dynamical heavy-ion collisions, although these encoders may not be intuitive. The deep

CNN provides a powerful and efficient decoder from which the EoS information can be extracted directly from the $\rho(p_T, \phi)$. It is in this sense that the high-level representations, which help decoding the EoS information in the present method, act as an EoS-meter for the QCD matter created in heavy-ion collisions. The Prediction Difference Analysis method is employed to extract the most relevant features for the classification task, which may inspire phenomenological and experimental studies. Our study might provide a key to the success of the experimental determination of QCD EoS and search for the critical end point. Another intriguing application of our framework is to extract the QGP transport coefficients from heavy-ion collisions. The present method can be further improved by including hadronic rescattering and detector efficiency corrections.

## Methods

**Network architecture.** The decisive ingredients for the success of hydrodynamic modeling of relativistic heavy-ion collisions are the bulk-matter EoS and the viscosity. In the study of the QCD transition in heavy-ion collisions, one of the holy-grail question is: how to reliably extract EoS and the nature of the QCD transition from the experimental data? The CNN[41,42] is a powerful technique in tasks such as image and video recognition, natural language processing. Supervised training of the CNN with labeled $\rho(p_T, \phi)$ generated by CLVisc is tested with $\rho(p_T, \phi)$ generated by iEBE-VISHNU. The training and testing $\rho(p_T, \phi)$ can be regarded as numerical experimental data. Hence, analyzing real experimental data is possible with straightforward generalizations of the current prototype setup.

Our CNN architecture is shown in Fig. 3. The input $\rho(p_T, \phi)$ consists of 15 $p_T$-bins and 48 $\phi$-bins. We use two convolutional layers each followed by batch normalization[43], dropout[44,45] with a rate 0.2 and PReLU activation[46]. These technical terms are briefly explained in Supplementary Note 1. In the first convolutional layer, there are 16 filters of size $8 \times 8$ scanning through the input $\rho(p_T, \phi)$ and creating 16 features of size $15 \times 48$. These features are further convoluted in the second convolutional layer that has 32 filters of size $7 \times 7 \times 16$. The weight matrix of both convolutional layers are initialized with normal distribution and constrained with L2 regularization[47]. In a convolutional layer, each neuron only locally connects to a small chunk of neurons in the previous layer by a convolution operation—this is a key reason for the success of the CNN architecture. Dropout, batch normalization, PReLU and L2 regularization work together to prevent overfitting that may generate model-dependent features from the training data set and thus hinder the generalizability of the method. The resulting 32 features of size $8 \times 24$ from the second convolutional layer are flattened and connected to a 128-neuron fully connected layer with batch normalization, dropout with rate 0.5 and sigmoid activation. The output layer is another fully connected layer with softmax activation and two neurons to indicate the type of the EoS. For multi-class classification, one may use more neurons in the output layer.

There are several non-trainable parameters in the neural network, such as the number of hidden layers, the size of the convolution kernels, the size of the final hidden layer and the dropout rate. The neural network in the present work can be easily rebuilt with these hyper-parameters in Keras[48] (the source code is also available as requested). These parameters are adjusted heuristically to maximize the

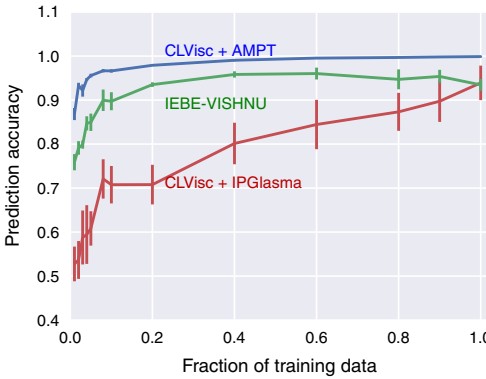

**Fig. 4** The dependence of testing accuracies on training data size. The prediction accuracy on testing data when different fractions of the training data is used to train the network. The solid lines and error bars represent the mean and the standard deviation of prediction accuracies from trained models in 10-fold cross validation method

training accuracy and validation accuracy but not the testing accuracy. The first step is to choose the number of hidden layers, the size of the convolution kernels and the size of the final hidden layer such that the model has enough capacity to describe the training data. At this step, we use a small portion of the training data, tune the widely used values of parameters and observe big training accuracy but small validation accuracy. It is found that the widely used convolution kernel sizes $5 \times 5$ and $3 \times 3$ do not work well at this step and increasing the number of the convolution layers from 2 to 3 does not improve the training accuracy and the validation accuracy. The next step is to increase the validation accuracy, in addition to the batch normalization and L2 regularization, it is found that dropout with a proper rate and tuning the size of the final hidden layer help to increase the validation accuracy. With this minimal working neural network, the validation accuracy increases rapidly with more training data. What is interesting is that when there are big training data, the previously not functioning architectures (with smaller convolution kernels and more hidden layers) also start to work and produces similar testing accuracy. The optimal neural network architecture and the values of the non-trainable parameters with big training data may desire future investigation.

**Training and validation**. We use supervised learning to tackle this binary classification problem with the crossover case labeled by $(1, 0)$ and the first-order case labeled by $(0, 1)$. The difference between the true label and the predicted label from the two output neurons, quantified by cross entropy[49], serves as the loss function $l$ $(\theta)$, where $\theta$ are the trainable parameters of the neural network. Training attempts to minimize the loss function by updating $\theta \to \theta - \delta\theta$. Here $\delta\theta = \alpha \, \partial l(\theta)/\partial\theta$ where $\alpha$ is the learning rate with initial value 0.0001 and adaptively changed in AdaMax method[50].

We build the architecture using Keras with a TensorFlow (r1.0)[51] backend and train the neural network with 2 NVIDIA GPUs K20m. The training data set is fed into the network in batches with batch size empirically selected as 64. One traversal of all the batches in the training data set is called one epoch. To accelerate the learning, the training data set is reshuffled before each epoch. The neural network is trained with 500 epochs. Small fluctuations of validation accuracy saturated around 99% are observed. The model parameters are saved to a new checkpoint whenever a smaller validation error is encountered.

The $k$-fold stratified cross validation is employed to estimate the model uncertainties. The training data set is randomly shuffled and split into $k$ equal folds with each fold containing equal number of two types of training data. One of these $k$ folds is used for validation while the other $k - 1$ folds are used for training. Finally $k$ models (according to $k$ pairs of (training, validation) partitioning) are trained to get the mean prediction accuracy and standard deviation. As shown in Fig. 4, the prediction accuracy approaches 99% with negligible uncertainty for testing on CLVisc + AMPT (same data generator as training), using less than 50% of the training data. While for the testing on IEBE-VISHNU + MC-Glauber (testing Group 1) and CLVisc + IP-Glasma (testing Group 2), the prediction accuracy increases as one increases the size of the training data set, which is in line with the practical expectation that more training data could boost the network's performance. With the full training data, we get on average a larger than 95% prediction accuracy, which is a very positive manifestation of the generalization capability of our deep CNN.

For the network settings, most of the parameters are introduced in the fully connected layers. In an alternative model, we add two more convolutional layers with filter size (3, 3) and subsequent average pooling layers to reduce the number of neurons in the flatten layer and also in the first fully connected layer, which helps to reduce the total number of parameters by a factor of 10. This deeper neural

network produces similar prediction accuracy and model uncertainty in a 10-fold cross validation tests.

The input images in the present method are particle density distributions in the momentum space. Due to collective expansion of the QGP, fluctuations in the initial state are transformed to strong correlations of final state particles in the images. These local structures and translational invariance of odd-order Fourier decomposition along the azimuthal angle direction make convolution neural networks preferable to fully connected neural networks.

The relativistic hydrodynamic simulations of the heavy ion collisions are quite computing intensive, even with the GPU parallelization, it still takes much longer to accumulate enough training data than the training process. In the beginning of this study when the training data size is not big enough, we experimented with fully connected neural networks. However, the network always overfits the training data and fails to work with the validating data. We noticed that CNN has much better generalizability than fully connected neural network with small set of data. With 22,000 events, the best performance of fully connected neural networks, with 2–5 hidden layers, gave on average 90% recognition rate on the testing data. Data augmentation in fully connected neural networks bring negligible improvement (less than 1%) on the testing data. The fully connected neural networks neglect the translation invariance of the local correlations of particles that are close to each other in momentum space.

**Data availability**. The data sets generated and analyzed during the current study are available in the public repository[52], https://doi.org/10.6084/m9.figshare.5457220.v1.

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

## Acknowledgements

L.-G.P. and H.P. acknowledge funding of a Helmholtz Young Investigator Group VH-NG-822 from the Helmholtz Association and the GSI Helmholtzzentrum für Schwerionenforschung (GSI). N.S. and K.Z. acknowledge the generous support of their DL-research at FIAS by SAMSON AG, Frankfurt and the support from GSI. H.St. acknowledges the support through the Judah M. Eisenberg Laureatus Chair at Goethe University. L.-G.P. and X.-N.W. are supported in part by the National Science Foundation (NSF) within the framework of the JETSCAPE collaboration, under grant number ACI-1550228. X.N.W was supported in part by NSFC under the Grant No. 11521064, by MOST of China under Grant No. 2014DFG02050, by the Major State Basic Research Development Program (MSBRD) in China under the Grant No. 2015CB856902and by U. S. DOE under Contract No. DE-AC02-05CH11231. This work was supported in part by the Helmholtz International Center for the Facility for Antiproton and Ion Research (HIC for FAIR) within the framework of the Landes-Offensive zur Entwicklung Wissenschaftlich-Oekonomischer Exzellenz (LOEWE) program launched by the State of Hesse. The computations were done in the Green-Cube GPU cluster LCSC at GSI, the Loewe-CSC at Goethe University, NERSC at LBNL and the GPU cluster at Central China Normal University.

## Author contributions

L.-G.P. contributed to the idea, the training and the second testing data set, the neural network construction for training/testing and the manuscript preparation; K.Z. contributed to the idea, the first testing data set, intensive discussions on neural network structures, physical explanations of the results and the manuscript edition; N.S. contributed to intensive discussions on neural network structures, physical explanations of the results and the manuscript edition; H.P., H.S. and X.-N.W. contributed to computing resources, physical insights and manuscript editions.

## Additional information

**Competing interests:** The authors declare no competing financial interests.

