## [Peer Review File · Nature Communications]

Reviewers' comments:

Reviewer #1 (Remarks to the Author):

In this manuscript, the authors develop a deep learning technique to study QCD bulk properties from heavy-ion observables. They use relativistic hydrodynamic models to simulate azimuthal particle spectra of charged pions. The deep learning is found to be successful in distinguishing the crossover and the first order transitions in the equation of state. The authors find that the result is not much affected by initial conditions, viscosity and freeze-out temperature. They conclude that the technique would become useful in future when searching for the critical point on the QCD phase diagram.

I think the manuscript is well written, but there are several issues which have to be addressed before it can be considered for publication. My individual comments are below:

1. The work reminds me of those by the MADAI collaboration. They have done semi-automated systematic analyses of the EoS [S. Pratt et al., Phys. Rev. Lett. 114, 202301 (2015)]. The paper is mentioned briefly in the manuscript, but the authors should clarify what is fundamentally new in terms of physics, why the results presented in the manuscript cannot be obtained by the previous method, etc.
2. In Sec. I, the authors state that "Thus far no noticeable and unique correspondence ... has been established using conventional observables." This sounds a little too much, since there have been a lot of study to obtain, for example, viscosity from heavy-ion data [M. Luzum and P. Romatschke, Phys. Rev. C78, 034915 (2008)].
3. Are the parameters in the two hydrodynamic models VISHNU and CLVisc tuned to reproduce the experimental data?
4. VISHNU has, so far as I know, UrQMD as a hadronic transport module whereas CLVisc does not. (The former stands for "Viscous Israel-Steward Hydrodynamics 'n' UrQMD".) Have the transport been switched off in VISHNU since resonance decay is not considered in the current work?
5. In Sec. II C, it would be helpful to have a mathematical definition of the prediction difference.
6. It is not clear to me how quantitatively different EOSL and EOSQ are. This can be important because the identification could have been made based on a structure of the EoS other than the phase transition. The authors could plot a thermodynamic quantity such as pressure as a function of temperature for the two EoS.
7. In Fig. 2, the difference between EOSL and EOSQ becomes less clear when shear viscosity is included. This seems to be reflected on the difference in the level of accuracy for Group 1 and 2 in Table III. It implies that if one selects most viscous events with $\eta/s = 0.10-0.16$ out of Group 1, the accuracy would change accordingly. The authors should discuss quantitatively how viscosity affects the accuracy in the manuscript.
8. Have the authors checked that Fig. 2 (b) reduces to (d) if viscosity is turned off in VISHNU?
9. In Fig. 2 (d), what is the physics behind the two-band structure? The one with $p_T \sim 0.3$ seems to contribute more to anisotropy than the one with $p_T \sim 0.8$ GeV. Do they correspond to the hadronic and the QGP phases?

10. Most experimental data, including particle spectra, are available in event-averaged forms. Can the proposed deep learning technique be used in that case? The authors should show how this is done.

Reviewer #2 (Remarks to the Author):

This manuscript presents an application of neural network machine learning methods to physics (QCD transition). Generally speaking the topic is interesting and timely.

There have been other recent successful applications of deep learning methods to other problems in physics (e.g. high energy physics). The experiments they consider produces data in the form of "images" and thus it is natural to use convolutional neural network to process these images. In short, the basic idea is sound and interesting. The manuscript, however, has a number of content and form weaknesses.

In terms of content, there are several methodological issues.

- 1) there is no comparison to other methods
- 2) there are no error bars on the result, for instance on the accuracy measures--also why not use full cross-validation/
- 3) there is no analysis of the relationship between the number of parameters of the models and the number of training examples,
- 4) there is no analysis of whether there is any overfitting or not
- 5) why didn't the authors produce more examples?
- 6) the data is simulated and not real data (although this is not a major fault--several other papers have published results with simulated data)
- 7) is the image processing problem being considered "translation invariant"? this point should be discussed
- 8) the authors could have used one of the pre-trained networks for computer vision that are freely available, remove the top layer, and retrain it on their data. This is a common approach these days and it would provide a baseline or a point of comparison.

In terms of form,

- 1) I am not sure it is a good idea to have "EoS" in the title. EoS is NOT a widely used acronym (Equation of State).
- 2) The word "formidable" in the abstract seems to strong given what is being presented.
- 3) Most experts would not agree with the definition of deep learning given in the first sentence of the introduction. This definition is too narrow.
- 4) Personally I did not find the description of the physics clear enough for a non-expert.
- 5) In contrast, the "tutorial" on deep learning in the Appendix is very elementary and does not seem belong to such a paper.
- 6) The approach is contrasted with the "Bayesian method which determines multiple properties a the same time". Predicting multiple things at the same time is something that neural networks can do very well (transfer learning etc). So it is not clear that this is a major difference.
- 7) Mixing mathematical symbols such as ">" with English text is probably not a good idea.
- 8) I cannot make any sense of the last sentence on page 3.
- 9) There are quite a few English mistakes or expressions that are awkward. A few examples:
"is generated by event-by-event hydrodynamic package CLVisc"
"The so-called convolutional neural network"--uding "so-called" is completely unnecessary
"We refer the readers to appendix"

Reviewer #3 (Remarks to the Author):

The paper "An EoS-meter of QCD transition from deep learning" applies recent advances in machine learning to the question of determining the parameters of the QCD transition from experimental data. With the wealth of experimental data at RHIC and LHC taken over the past 17 years, the field has recognized that to make progress in determining the physical characteristics of QCD matter created in heavy ion collisions one must fold as much data as possible into as sophisticated a model as possible. This is not a simple task, as the connection between the physical parameters and measurements is not directly one-to-one. This manuscript explores one method to handle the wealth of data, through machine learning, among others that are being developed, most significantly the Bayesian techniques.

The paper provides a reasonably convincing demonstration of the power of the deep learning technique on a set of models with widely differing parametric conditions, returning the correct parameters with high accuracy. I say "reasonably" because important aspects in assessing the models against experimental data (multiple species of hadrons besides pions, resonance decays, experimental limitations, etc.) are not included in the analysis. The final paragraph in section II b touches on these limitations. The devil in many historical uses of complicated algorithms for analysis of actual data has been in the details.

That said, this manuscript reports an important first step in the use of these techniques, which I am confident will spark follow-up with more detailed comparisons to real experimental data, and I recommend its publication.

Reviewer 1

Referee’s comment: *Reviewer 1 (Remarks to the Author): In this manuscript, the authors develop a deep learning technique to study QCD bulk properties from heavy-ion observables. They use relativistic hydrodynamic models to simulate azimuthal particle spectra of charged pions. The deep learning is found to be successful in distinguishing the crossover and the first order transitions in the equation of state. The authors find that the result is not much affected by initial conditions, viscosity and freeze-out temperature. They conclude that the technique would become useful in future when searching for the critical point on the QCD phase diagram. I think the manuscript is well written, but there are several issues which have to be addressed before it can be considered for publication.*

Authors’ reply: We thank the referee for his/her carefully reading and thoughtful comments. Please find our replies to individual questions below.

Referee’s comment: *The work reminds me of those by the MADAI collaboration. They have done semi-automated systematic analyses of the EoS [S. Pratt et al., Phys. Rev. Lett. 114, 202301 (2015)]. The paper is mentioned briefly in the manuscript, but the authors should clarify what is fundamentally new in terms of physics, why the results presented in the manuscript cannot be obtained by the previous method, etc.*

Authors’ reply: The Bayesian method used by the MADAI collaboration and the recent global fitting by the Duke group are great attempts to introduce the statistical machine learning technique with Bayesian method to heavy ion physics. These studies aim to constrain model parameters by calibrating them to the experimental data on a predefined set of event-averaged observables. In contrast, the machine learning technique with deep convolutional neural networks (DCNNs) employed in this study uses event-by-event raw data instead of predefined event-averaged quantities. This way, the DCNNs can find hidden correlations that are sensitive to physical properties of the system. In our exploratory study with this novel method, we demonstrate, on the one hand, the existence of discriminative and traceable projections – “encoders” – from the QCD transition onto the final-state particle spectra $\rho(p_T, \Phi)$. On the other hand, by employing DCNNs, we build a powerful and efficient “decoder” from which the exclusive response of the EoS can be extracted directly from the particle spectra $\rho(p_T, \Phi)$. In the future, we would like to combine the deep learning and Bayesian methods together to tackle challenges in physics and related areas.

We have modified part of the second and third paragraph in the introduction to point to these differences (see the following)

Referee’s comment: *In Sec. I, the authors state that ”Thus far no noticeable and unique correspondence ? has been established using conventional observables.” This sounds a little too much, since there have been a lot of study to obtain, for example, viscosity from heavy-ion data [M. Luzum and P. Romatschke, Phys. Rev. C78, 034915 (2008)].*

Authors' reply: We agree with the referee that our statement here might be too strong and unclear. This point is also connected to comparison to the Bayesian method by the MADAI collaboration. We have modified these statements to the following.

"Current efforts to extract physical properties of the QCD matter from experimental data are through direct comparisons with model calculations of event-averaged and predefined observables such as anisotropic flow [18] or global fitting of a set of observables with Bayesian method [28,29]. However, event-by-event raw data on $\rho(p_T, \Phi)$ at different rapidities provide much more information that contains hidden correlations. These hidden correlations can be sensitive to physical properties of the system but independent of other model parameters."

Together with the modified third paragraph, the revised statement will acknowledge both the achievements and short-comings of conventional methods, and how our DL method is different.

Referee's comment: *Are the parameters in the two hydrodynamic models VISHNU and CLVisc tuned to reproduce the experimental data?*

Authors' reply: Both VISHNU and CLVisc with corresponding initial state conditions can reproduce the experimental data with tuned values of parameters such as viscosity, initial time and freeze-out temperature. The aim of our study here is to find out exclusive hydrodynamic responses to the EoS, which are general features of the QCD transition in a fluid system. These features are robust against model parameters, and can help us to extract physical information from raw experimental data. For this purpose, we have not and don't necessarily need to require all sets of parameters to reproduce the experimental data.

Referee's comment: *VISHNU has, so far as I know, UrQMD as a hadronic transport module whereas CLVisc does not. (The former stands for "Viscous Israel-Steward Hydrodynamics 'n' UrQMD".) Have the transport been switched off in VISHNU since resonance decay is not considered in the current work?*

Authors' reply: Yes, we have switched off the transport module in VISHNU in the current work. The following statement has been added in the revised manuscript.

"the hadronic transport module UrQMD is switched off in iEBE-VISHNU to exclude contributions from resonance decays in testing data".

Referee's comment: *In Sec. II C, it would be helpful to have a mathematical definition of the prediction difference.*

Authors' reply: Done. The mathematical definition is appended to the supplementary material.

Referee's comment: *It is not clear to me how quantitatively different EOSL and EOSQ are. This can be important because the identification could have been made based on a structure of the EoS other than the phase transition. The authors could plot a thermodynamic quantity such as pressure as a function of temperature for the two EoS.*

Authors' reply: The difference between the widely used EOSL and EOSQ is shown in the small chart of the phase diagram, where we plotted the pressure as a function of energy density. The EOSQ differs from EOSL not only around the phase transition region but also at higher temperatures. In order to prove that the network really identifies the type of the QCD transition, we have composed one modified EOSQ whose EoS is replaced with EOSL for energy densities outside the first order phase transition region $[0.5, 1.9] \text{ GeV}/fm^3$. With this modified EOSQ, we find that 199 out of 200 events are classified to EOSQ, for Au+Au $\sqrt{s_{NN}} = 200 \text{ GeV}$ collisions with CLVisc+IPGlasma initial condition. In principle, we can also use this modified EOSQ for training to fully prove that the network identifies the nature of QCD transition instead of other differences in the EoS.

Referee's comment: *In Fig. 2, the difference between EOSL and EOSQ becomes less clear when shear viscosity is included. This seems to be reflected on the difference in the level of accuracy for Group 1 and 2 in Table III. It implies that if one selects most viscous events with $\eta/s = 0.10-0.16$ out of Group 1, the accuracy would change accordingly. The authors should discuss quantitatively how viscosity affects the accuracy in the manuscript.*

Authors' reply: This is a very good point. We have checked that as the shear viscosity over entropy density ratio η/s increases, the mean prediction accuracy decreases from 97.1% (for $\eta/s = 0.0$) to 96.6% (for $\eta/s = 0.08$) and 87% (for $\eta/s = 0.16$) in a 10-fold cross validation tests. However, we can not make a strong statement saying that increasing η/s will decrease prediction accuracy. The reason is that we have not included $\eta/s > 0.08$ in our training dataset.

We added the following discussion on this issue in the revised manuscript:

The mean prediction accuracy decreases from 97.1% (for $\eta/s = 0.0$) to 96.6% (for $\eta/s = 0.08$) and 87% (for $\eta/s = 0.16$) in the 10-fold cross validation for testing GROUP 1.

Referee's comment: *Have the authors checked that Fig. 2 (b) reduces to (d) if viscosity is turned off in VISHNU?*

Authors' reply: In our analysis, the prediction difference analysis method is applied to each event with a specific model setup (initial conditions, model parameters and PDE solver). The importance map shown in Fig. 2 is event-averaged for different model setups and values of the shear viscosity. The difference between Fig. 2 (b) and (d) (in the old version) is in fact caused by both different values of viscosity and model setup (Group 1: IEBC-VISHNU+MC-Glauber, Group 2: CLVisc+IP-Glasma). To isolate the influence of viscosity and model setup separately, we have carried out additional calculations. In the revised Fig. 2, we compare not only the importance maps for different EoS (EOSL in the first versus EOSQ in the second row for the same model setup, but also for the same model setup with different viscosity (first versus the second column) and for different model setup with the same viscosity (first versus the third column). We can see that for the same EoS, model setup and the value of shear viscosity both influence the importance maps. However, the importance

maps for EOSL in general have a distinctly narrower width in the p_T range than that for EOSQ, independently of the model setup and the value of viscosity.

We have modified the discussion about the revised Fig. 2 accordingly as follows:

Shown in Fig. 2 are importance maps which illustrate the (p_T, Φ) dependence of the mean prediction difference averaged over 800 events for different model setups (initial conditions, PDE solver and model parameters), EoSs and values of the shear viscosity. For a given event, the mean prediction difference in each (p_T, Φ) bin is computed against 10 random reference events from the same dataset. Comparing different columns in the same row in Fig. 2, we can see that importance maps vary slightly for different values of viscosity and model setup (Group 1: IEBE-VISHNU+MC-Glauber, Group 2: CLVics+IP-Glasma) for the same EoS. However, importance maps for EOSL in general have a distinctly narrower width in the p_T range than that for EOSQ, independently of the model setup and the value of viscosity. This might be the important region of hidden features the network recognizes in classifying the EoS under each event.

Referee's comment: *In Fig. 2 (d), what is the physics behind the two-band structure? The one with $p_T \sim 0.3$ seems to contribute more to anisotropy than the one with $p_T \sim 0.8$ GeV. Do they correspond to the hadronic and the QGP phases?*

Authors' reply: In our revised Fig. 2 with different EOSs, model setups and values of shear viscosity, the two-band feature only appears in some of the configurations. Due to the complexity of the learnt functions by the DCNN, it is difficult to understand the physics mechanism underlying the structures in the importance map. This is still a challenge in the community.

Referee's comment: *Most experimental data, including particle spectra, are available in event-averaged forms. Can the proposed deep learning technique be used in that case? The authors should show how this is done.*

Authors' reply: Deep learning utilizes vast amount of data containing many raw information, including event-to-event fluctuations, to train the network and recognize some hidden correlations. This is especially true for the DCNN's. Event-averaging will inevitably leads to the loss of these hidden correlations. One can, however, reduce the dimension of the experimental data. Our current network utilize the particle spectra in one slice of (or integrated over) rapidity. One can in principle integrate out the azimuthal angle and use event-by-event distributions of p_T spectra to develop a one dimensional convolution neural network to capture the hadron correlations in p_T .

Reviewer 2

Referee's comment: *Reviewer 2 (Remarks to the Author): This manuscript presents an application of neural network machine learning methods to physics (QCD transition). Generally speaking the topic is interesting and timely. There*

have been other recent successful applications of deep learning methods to other problems in physics (e.g. high energy physics). The experiments they consider produces data in the form of "images" and thus it is natural to use convolutional neural network to process these images. In short, the basic idea is sound and interesting. The manuscript, however, has a number of content and form weaknesses.

Authors' reply: We thank the referee for carefully reading and thoughtful comments. Please find our replies to individual questions below.

Referee's comment: *there is no comparison to other methods*

Authors' reply: As the referee points out, there are several applications of machines learning to other problems in physics. As far as we know, our study is the first application to the heavy-ion physics using the hadron spectra as "images". There are other methods for extracting physics from pre-defined and event averaged observables with direct model comparison or a set of observables with Bayesian method. We have revised the introduction to acknowledge these methods.

Referee's comment: *there are no error bars on the result, for instance on the accuracy measures—also why not use full cross-validation/*

Authors' reply: It is a very good suggestion to use cross-validation to estimate the model error. We tried 10-fold stratified cross-validation for the 40000 events given by the first Monte Carlo model (CLVisc+AMPT initial condition), and use 4000 for testing. The accuracy is 99.88% (+/- 0.04%) where the model uncertainty is 0.04% which is very small. These 10 pre-trained models are used to compute the mean and the standard deviation of prediction accuracy on another 2 groups of testing datasets. For the first group (IEBE-VISHNU+MC-Glauber initial condition), the testing accuracy is 93.46% (+/- 1.35%), for the second group (CLVisc+IP-Glasma initial condition), the testing accuracy is 95.12% (+/- 3.08%).

We have added the following content in the revised manuscript:

(1) The mean prediction accuracy and model uncertainties from cross validation method are added to Table III.

(2) The prediction accuracy and model uncertainties as a function of the training data size are added in Fig. 4 with corresponding discussions.

Referee's comment: *there is no analysis of the relationship between the number of parameters of the models and the number of training examples,*

Authors' reply: In deep learning studies, the effective number of parameters is unknown before training, especially with L1, L2 regularization applied. In the handwritten digit recognition problem MNIST, even with a fully connected neural network that takes much more parameters than training examples, the model still produce reasonable generalization capability. In the fine tuning tutorial "Building powerful image classification models using very little data" by Francois Chollet, there are only 2000 training examples for images of dogs and

cats with input shape (150, 150, 3), with a simple convolution neural network (not the fine tuning one) with structure conv2d(32, 3, 3) + maxpooling(2, 2) + conv2d(32, 3, 3) + maxpooling(2, 2) + conv2d(64, 3, 3) + maxpooling(2, 2) + flatten + fc(64) takes approximately 1.4 million number of parameters, which is around 700 times more than the number of training examples, however, that model still have prediction accuracy around 71 – 81% on the testing dataset. In our present method, the number of parameters with dropout is about 10 times more than the number of training examples. We have also tried to reduce the number of parameters by stacking more hidden layers with less neurons in each layer, in a recent test, we use conv2d(16, 5, 5) + conv2d(32, 5, 5) + avgpooling(2, 2) + conv2d(32, 3, 3) + avgpooling(2, 2) + conv2d(32, 3, 3) + avgpooling(2, 2) + flatten + fc(64) + fc(2) structure which reduces the number of parameters to the same level as the training examples. In the 10-fold cross validation test, the mean prediction accuracy is similar to our current model and the prediction variance is improved slightly. As a comparison, we listed the results from this new neural network:

Data generator	Mean testing accuracy	std
CLVisc + AMPT	0.999	0.03%
IEBE-VISHNU	0.952	0.8%
CLVisc + IP-Glasma	0.963	2.1%

Table 1: Prediction accuracy and model uncertainties with a deeper and narrower convolution neural network.

Referee’s comment: *there is no analysis of whether there is any overfitting or not*

Authors’ reply: We have checked overfitting rigorously during the research. First, the data are split into 3 parts: training data, validation data and testing data. The testing data has never been used for training. When we compute the prediction accuracy, we always use testing data. In the testing data, we even checked the model (generator) dependence by employing a different Monte Carlo model with different fluctuating initial conditions, different eta/s and freeze out temperature. In our convolution neural network with dropout, the prediction accuracy on testing data is always increasing as we increase the training dataset. In the revised manuscript, the prediction accuracy as a function of the size of the training dataset is shown in Fig.4 and discussions are also added.

Referee’s comment: *why didn’t the authors produce more examples?*

Authors’ reply: In principle, the validation error and testing error decrease as the training dataset size increases. Preparing data (solving relativistic hydrodynamics with shear viscosity) is time consuming and we decided to stop producing more data when the validation error becomes low enough (less than 1%). We have compared the size of our training dataset with the MNIST handwritten digits recognition problem, we used 40000 examples (720 pixels in each

image) for 2 classes while MNIST uses 65000 examples (784 pixels in each image) for 10 classes. We believe that we have enough training examples for each class. The prediction accuracy as a function of training data size is shown in the updated manuscript.

Referee's comment: *the data is simulated and not real data (although this is not a major fault—several other papers have published results with simulated data)*

Authors' reply: Applying machine learning to real data is our ultimate goal. However, currently there is not much raw data available from experiments, which is in general not accessible to non-members of the experiment. With studies like ours, we are spreading the idea to use more data (either designed observables or raw data) in the deep learning study and also in the Bayesian analysis. All data currently available for the public from heavy ion experiments are preprocessed by reducing to very low dimensions and averaged over many events. The data from feature engineering might lose sensitive information when we want to extract exclusive dynamical properties. Hopefully, the situation will change and more raw data will be released in the near future.

Referee's comment: *is the image processing problem being considered "translation invariant"? this point should be discussed*

Authors' reply: The answer is yes. On one hand, for the problem we are targeting (using final spectra $\rho(p_T, \Phi)$ to detect EoS classes in a heavy-ion collision), it's invariant under the shift along horizontal (Φ) axis for the spectra since only the relative angle between particles in a collision event matters. On the other hand, from the robust performance of the deep CNN on testing dataset from different (hydro-)models, and especially the different importance maps for group 1 (IEBE-VISHNU+MC-Glauber) and group 2 (CLVisc+IP-Glasma) testing data that been recognised by the network, it can be seen that the deep CNN behaves in an invariant way to find out the relevant correlation from different individual input.

Referee's comment: *the authors could have used one of the pre-trained networks for computer vision that are freely available, remove the top layer, and retrain it on their data. This is a common approach these days and it would provide a baseline or a point of comparison.*

Authors' reply: In the problem we are studying, we treat the final particle spectra from heavy-ion collisions as 'images' but not exactly in the same sense of an image as in the freely available networks. The configuration space (the spectra 'image' is represented in momentum space while the usual image is represented in position space) and feature space in our study are different. So it's hard to directly use pre-trained models which mostly aim to identify real objects in position configuration. In addition, most of the pre-trained models use 3 color channels while in our current case only 1 channel is considered.

Referee's comment: *I am not sure it is a good idea to have "EoS" in the title. EoS is NOT a widely used acronym (Equation of State).*

Authors' reply: Thanks for the good suggestion. We have changed it to "equation-of-state-meter" in the revised manuscript.

Referee's comment: *The word "formidable" in the abstract seems too strong given what is being presented.*

Authors' reply: We have changed it to 'powerful' in the revised manuscript.

Referee's comment: *Most experts would not agree with the definition of deep learning given in the first sentence of the introduction. This definition is too narrow.*

Authors' reply: In the revised manuscript, we have changed the description of deep learning to a more general one:

Deep learning (DL) is a branch of machine learning that learns multiple levels of representations from data [1,2].

Referee's comment: *Personally I did not find the description of the physics clear enough for a non-expert.*

Authors' reply: We have improved the description of the physics in the revised introduction:

Strong interaction in nuclear matter is governed by the theory of Quantum Chromodynamics (QCD). It predicts a transition from the normal nuclear matter, in which the more fundamental constituents, quarks and gluons, are confined within the domains of nucleons, to a new form of matter with freely roaming quarks and gluons as one increases the temperature or density. The QCD transition is conjectured to be a crossover at small density (and moderately high temperature), and first order at moderate density (and lower temperature), with a critical point separating the two, see Fig. 1 for a schematic QCD phase diagram and [15-17] for some reviews. One primary goal of ultra-relativistic heavy-ion collisions is to study the QCD transition. Though it is believed that strongly coupled QCD matter can be formed in heavy-ion collisions at the Relativistic Heavy Ion Collider (RHIC, Brookhaven National Laboratory, USA), Large Hadron Collider (LHC, European Organization for Nuclear Research, Switzerland), and at the forthcoming Facility for Anti-proton and Ion Research (FAIR, GSI Helmholtz Centre for Heavy Ion Research, Germany), a direct access to the bulk properties of the matter such as the equation of state (EoS) and transport coefficients is impossible due to the highly dynamical nature of the collisions. In heavy-ion collisions where two high-energy nuclei collide along the longitudinal (z) direction, what experiments measure directly are the final-state particle distributions in longitudinal momentum (rapidity), transverse momentum p_T and azimuthal angle Φ . Current efforts to extract physical properties of the QCD matter from experimental data are through direct comparisons with model calculations of event-averaged and predefined observables such as anisotropic flow [18] or global fitting of a set of observables with Bayesian method [28,29]. However, event-by-event raw data on $\rho(p_T, \Phi)$ at different rapidities provide much more information that contains hidden correlations. These hidden correlations

can be sensitive to physical properties of the system but independent of other model parameters.”

Referee’s comment: *In contrast, the ”tutorial” on deep learning in the Appendix is very elementary and does not seem belong to such a paper.*

Authors’ reply: Since this is the first application of deep learning in relativistic heavy ion collision, we wish to explain the technical terms to readers in heavy ion community as well, who are not very familiar with deep learning. In our revised manuscript, this part will be shown in the supplementary material for non-machine learning experts.

Referee’s comment: *The approach is contrasted with the ”Bayesian method which determines multiple properties a the same time”. Predicting multiple things at the same time is something that neural networks can do very well (transfer learning etc). So it is not clear that this is a major difference.*

Authors’ reply: Our purpose here is to discuss the differences between the existing Bayesian method employed in heavy ion physics and the convolution neural network method we have developed in the present paper. We agree with the referee on this point and have changed the discussion accordingly in the second paragraph of the Introduction in the revised manuscript.

Referee’s comment: *Mixing mathematical symbols such as ”>” with English text is probably not a good idea.*

Authors’ reply: We have changed $>$ to English text ”larger than” in the revised manuscript.

Referee’s comment: *I cannot make any sense of the last sentence on page 3.*

Authors’ reply: We have changed the last sentence and the proceeding one to the following:

It has been recently pointed out that model dependent-features (features in the training data that depends on the simulation models and parameters) may generate large uncertainties in the network performance [6]. The network we develop below is, however, not sensitive to these model-dependent features.

Referee’s comment: *There are quite a few English mistakes or expressions that are awkward. A few examples: ”is generated by event-by-event hydrodynamic package CLVisc” ”The so-called convolutional neural network”–uding ”so-called” is completely unnecessary ”We refer the readers to appendix”*

Authors’ reply: We have revised our manuscript:

”generated by event-by-event hydrodynamic package CLVisc” is changed to ”generated from the relativistic hydrodynamic program CLVisc”.

”so-called” is removed in the revised manuscript.

”We refer the readers to appendix ...” is changed to ”These technical terms are briefly explained in the supplementary materials.”

Reviewer 3

Referee’s comment: *Reviewer 3 (Remarks to the Author): The paper "An EoS-meter of QCD transition from deep learning" applies recent advances in machine learning to the question of determining the parameters of the QCD transition from experimental data. With the wealth of experimental data at RHIC and LHC taken over the past 17 years, the field has recognized that to make progress in determining the physical characteristics of QCD matter created in heavy ion collisions one must fold as much data as possible into as sophisticated a model as possible. This is not a simple task, as the connection between the physical parameters and measurements is not directly one-to-one. This manuscript explores one method to handle the wealth of data, through machine learning, among others that are being developed, most significantly the Bayesian techniques. The paper provides a reasonably convincing demonstration of the power of the deep learning technique on a set of models with widely differing parametric conditions, returning the correct parameters with high accuracy. I say "reasonably" because important aspects in assessing the models against experimental data (multiple species of hadrons besides pions, resonance decays, experimental limitations, etc.) are not included in the analysis. The final paragraph in section II b touches on these limitations. The devil in many historical uses of complicated algorithms for analysis of actual data has been in the details. That said, this manuscript reports an important first step in the use of these techniques, which I am confident will spark follow-up with more detailed comparisons to real experimental data, and I recommend its publication.*

Authors’ reply: We thank the referee for the very positive report. Indeed, we are only using a small portion of data for this proof of principle study. There is much more wealth hidden in the experimental data at RHIC and LHC and the event-by-event Monte Carlo simulations. The modern machine learning techniques and big data analysis have the potential to explore these data.

Additional changes to the manuscript

Change 1) The abstract is slightly simplified,

Supervised learning with a deep convolutional neural network is used to identify the QCD equation of state (EoS) employed in relativistic hydrodynamic simulations of heavy-ion collisions. ~~The~~ **from the simulated** final-state particle spectra $\rho(p_T, \Phi)$ ~~provide directly accessible information from experiments.~~ High-level correlations of $\rho(p_T, \Phi)$ learned by the neural network act as an **effective** "EoS-meter", ~~effective~~ in detecting the nature of the QCD ~~phase~~ transition.

Change 2) The optimal prediction accuracy shown in Table III for 2 groups of testing data is replaced with mean prediction accuracy together with model uncertainties for 3 groups of testing data, given by 10-fold cross validation method.

Change 3) We have removed "We randomly select 10% of all the $\rho(p_T, \Phi)$ for validation and use the rest for training." in sec. II.A Training and testing datasets, since the 10-fold cross validation method is leveraged in the revised manuscript and the new implementations are described in details in the Method section.

Change 4) In the revised manuscript, we pick out 4000 events from the training dataset and use them as testing GROUP 0. We have added the following sentence in sec. II.A,

We pick out 4000 events randomly and use them as the testing dataset GROUP 0. The remaining 40000 events are used for training and validation.

Change 5) We have added the 10-fold cross validation tests for different training data sizes and corresponding discussions in the revised manuscript.

Change 6) The L1 and L2 regularization terms θ and θ^2 in the supplementary material are replaced with a more formal definition as follows,

$$L1 : l(\theta) = l(\theta) + \lambda \|\theta\|_1 \quad (1)$$

$$L2 : l(\theta) = l(\theta) + \lambda \|\theta\|_2^2 \quad (2)$$

where λ is the regularization strength, $\|\theta\|_p \equiv \left(\sum_j^n |\theta_j|^p\right)^{1/p}$ is the p -norm of the parameters $\theta = (\theta_1, \theta_2, \dots, \theta_n)$.

Change 7) The mathematical definition of prediction difference analysis is appended in the supplementary material

Prediction Difference Analysis is a method to visualize the difference between the log-odds of the prediction probability $p(y|\rho)$ and $p(y|\rho_{\setminus i})$, where y is the class

value, ρ is the real image and $\rho_{\setminus i}$ is the imperfect image without the knowledge of the i th pixel. The prediction difference is dubbed as *weight of evidence* [4,5].

$$\text{WE}_i(y|\rho) = \log_2(\text{odds}(y|\rho)) - \log_2(\text{odds}(y|\rho_{\setminus i})) , \quad (3)$$

where $\text{odds}(z) = p(z)/(1-p(z))$ is used to symmetrize $\log_2 p$ and $-\log_2(1-p)$, with Laplace correction $p \leftarrow (pn+1)/(n+m)$ to avoid zero probability, where n is the number of training instances and m is the number of classes. The $p(y|\rho_{\setminus i})$ is approximated by,

$$p(y|\rho_{\setminus i}) \approx \sum_s^{m_i} p(\rho_i) p(y|\rho \leftarrow \rho_i = a_s) , \quad (4)$$

with the i th pixel replaced with all the possible values a_s weighted by its value probability. The importance map is given by the mean weight of evidence over many events that have the same class label.

Change 7) We have added two more references,

`\bibitem{Luzum:2008cw}`

M.~Luzum and P.~Romatschke,

%'Conformal Relativistic Viscous Hydrodynamics:

Applications to RHIC results at $s(\text{NN})^{1/2} = 200\text{-GeV}$,''

Phys.\ Rev.\ C {\bf 78}, 034915 (2008).

%[arXiv:0804.4015 [nucl-th]].

`\bibitem{robnik2008explaining}`

M.~Robnik-Sikonja and I.~Kononenko,

%'Explaining classifications for individual instances,"

Knowledge and Data Engineering, IEEE Transactions on, 20(5):589-600, (2008).

Reviewers' comments:

Reviewer #1 (Remarks to the Author):

The authors have addressed most of the issues I raised. They have managed to explain how their deep learning method is unique in comparison to the conventional methods. The authors have also shown that VISHNU is handled properly in their work and that their method can recognize the type of QCD transition in an EoS. The effect of viscosity on the mean prediction accuracy is now studied quantitatively. The authors have also given an adequate explanation in regard to the application to the analyses of actual experimental data.

Physical interpretations of the importance maps, it seems to me, remain non-trivial, but given the level of the accuracy of EoS identification, it would be safe to say the deep learning method captures important aspects of the underlying physics. My optional suggestion is that the authors also present in Fig. 2 the map for EOS L and EOSQ with $\eta/s = 0.08$ for GROUP 2 for better illustration.

Overall, I believe the improved manuscript is in a good shape. Once the authors consider the optional point, I recommend its publication in Nature Communications.

Reviewer #4 (Remarks to the Author):

This paper constructs a classifier to distinguish between two equations of state, based on simulated data. The assertion of the paper is that the construction of such a classifier gives insights into the underlying phenomena. I am not a physicist and so I am not able to evaluate the value of such a classifier on its own or whether the insights claimed in the paper are of value.

However, I am an expert in deep learning and from a methodological point of view, this paper is weak. Fundamentally, deep learning is the construction of function approximators from example data, and constructing them in such a way that composition leads to useful features. The fact that the function in this case can be approximated is not, in of itself, an interesting scientific result. Many seemingly-arbitrary decisions have been made in the construction of the classifier -- architecture, nonlinearities, regularization, etc -- and it is unclear how these decisions were made and how much they impact the result. A much more rigorous evaluation is necessary because the core question should be "what is the simplest classifier that can successfully distinguish these EoS?" Are there very basic statistics of the data that give the same insights? What does a linear classifier do? What does a simple class-conditional Gaussian density reveal? Building a large black box function approximator with many knobs does not tell us much about the structure of the data. Indeed, I find it highly suspect to use a convolutional model in this case. Is it really the case that there is translation invariance in both axes? The response to reviewers claim there is invariance to Φ , but Figure 2 seems to contradict this. At a basic level, why is convolution even necessary? It provides a more compact parameterization, but the "images" here are tiny and an arbitrary amount of data is available, so why not train a fully-connected network and evaluate how it compares? This is just one example of how such a paper needs to peel back the layers of modeling complexity to try to identify what is providing information about the classification. How can we feel comfortable that the classification is not simply rising from some very simple first or second order statistics in the data? Deep learning methods are only interesting when other simpler things fail.

To summarize, I feel that much more empirical rigor is needed here if this work is intended to inspire

further experimental studies. In my opinion it is not sufficient to simply show that there exists a complicated function approximator to this problem; rather, one must tease apart the characteristics of failed and successful such approximators to reveal insights into the underlying phenomena.

Reviewer 1

Referee's comment: *Reviewer 1 (Remarks to the Author): The authors have addressed most of the issues I raised. They have managed to explain how their deep learning method is unique in comparison to the conventional methods. The authors have also shown that VISHNU is handled properly in their work and that their method can recognize the type of QCD transition in an EoS. The effect of viscosity on the mean prediction accuracy is now studied quantitatively. The authors have also given an adequate explanation in regard to the application to the analyses of actual experimental data. Physical interpretations of the importance maps, it seems to me, remain non-trivial, but given the level of the accuracy of EoS identification, it would be safe to say the deep learning method captures important aspects of the underlying physics. My optional suggestion is that the authors also present in Fig. 2 the map for EOS L and EOSQ with $\eta/s = 0.08$ for GROUP 2 for better illustration. Overall, I believe the improved manuscript is in a good shape. Once the authors consider the optional point, I recommend its publication in Nature Communications.*

Authors' reply: We thank the referee for positive comments. We have done relativistic hydrodynamic simulations to accumulate more data for EOSL and EOSQ with $\eta/s = 0.08$, the additional dataset is appended to GROUP2. For better illustration, we have followed the referee's suggestion to add the importance map for the new data. In the new version, we have made several changes w.r.t the new dataset. The conclusions are not changed. The data size has been adjusted for GROUP 2; the prediction accuracy and model uncertainty from cross validation method have been updated for GROUP 2.

Reviewer 2

Referee's comment: *This paper constructs a classifier to distinguish between two equations of state, based on simulated data. The assertion of the paper is that the construction of such a classifier gives insights into the underlying phenomena. I am not a physicist and so I am not able to evaluate the value of such a classifier on its own or whether the insights claimed in the paper are of value.*

Authors' reply: The study of the QCD phase structure and the search for the critical end point of the QCD transition are the central motivations of high-energy heavy-ion physics, which is also the only possibility in the laboratory on earth to study the very first instant after the birth of our Universe. Large international collaborations, both in theory and in experiment, have been searching for signals of this phase structure at huge accelerator centers worldwide. The beam energy scan project at the relativistic heavy-ion collider (RHIC) at Brookhaven National Laboratory in the US aims at locating the critical end point in the QCD phase diagram. This critical end point separates the crossover

transition and the conjectured first-order phase transition from hadrons to deconfined quark-gluon matter. Critical fluctuations are used in experiments to locate this critical end point. However, currently the observed signals are too weak to pin down its location. The present work proves that nonlinear classifiers – tools from deep learning, on the other hand, do successfully discriminate the two types of the QCD transition. Our finding is unprecedented and of key importance for further development in this research field.

Referee’s comment: *However, I am an expert in deep learning and from a methodological point of view, this paper is weak. Fundamentally, deep learning is the construction of function approximators from example data, and constructing them in such a way that composition leads to useful features. The fact that the function in this case can be approximated is not, in of itself, an interesting scientific result.*

Authors’ reply: We respectfully disagree with the referee on the scientific results in our study here. Our focus of this paper is indeed not on the methodology of deep learning itself, but on the application of ML to the field of high-energy heavy-ion physics and possibly a break-through in new analysis of future data to identify the nature of QCD phase transition, which is one of the most important physics questions in the century.

The functional approximation by deep learning might be trivial to some in the DL community, but in practice it still took many deep learning experts decades to realize it in practice such as mapping images to their labels, which brought revolution in computer vision and AI researches. Applying deep learning to physics problems is still in its nascent stage. A ‘simple’ fact (ML to be used as a tool) in one field (deep learning) can be of tremendous importance in another (high-energy heavy-ion physics and cosmology).

The important question we try to answer in this study is “is there a traceable encoder of the dynamical information from phase structure (EoS) that survives the evolution and exists in the final snapshot”, and if “yes”, then “how to exclusively and effectively decode these information from the highly complex final output”. These questions are crucial but unclear for decades in high-energy heavy-ion physics (and also in physical cosmology) due to the complexity and highly-dynamical characteristics in the collision evolution. For the first time our results demonstrate with the help of deep learning that exclusive and traceable encoder of information from QCD dynamical transition do exist inside the final emitted particle’s spectra. This opens a novel way in the field of high-energy heavy-ion physics to decode physics from complex data by using deep learning tools.

We have add the following to the discussion section,

“Beside the deep CNN method employed in the present paper, there are also some other machine learning classifiers. In the supplementary materials we attached the results from several traditional machine learning methods, such as support vector machine classifier (SVC), decision trees, random forests and gradient boosting trees. The best classifier (linear SVC) that generalizes well on two testing datasets achieves on average $\sim 80\%$ prediction accuracy. The

important features from different classifiers differ from each other, however, those with good generalization capability have similar importance regions as given by the deep CNN. The deep CNN with on average $\sim 95\%$ prediction accuracy works much better to answer the core questions – “Is there a traceable encoder of the dynamical information from phase structure (EoS) that survives the evolution and exists in the final snapshot?” and if “yes”, then “how to exclusively and effectively decode these information from the highly complex final output”. These questions are crucial but unclear for decades in high-energy heavy-ion physics (and also in physical cosmology) due to the complexity and highly-dynamical characteristics in the collision evolution. The deep CNN demonstrates the revolution that big data analysis and machine learning might bring to the high energy physics and astrophysics.”

Referee’s comment: *Many seemingly-arbitrary decisions have been made in the construction of the classifier – architecture, nonlinearities, regularization, etc – and it is unclear how these decisions were made and how much they impact the result. A much more rigorous evaluation is necessary because the core question should be “what is the simplest classifier that can successfully distinguish these EoS?” Are there very basic statistics of the data that give the same insights? What does a linear classifier do? What does a simple class-conditional Gaussian density reveal? Building a large black box function approximator with many knobs does not tell us much about the structure of the data.*

Authors’ reply: Firstly, as we explained in above, in the present paper we are focusing on physics – physical signal, physical information – but not the technique itself or points of view in data science. The core question in our field here and also discussed in our present work is “the existence of traceable encoder of the dynamical information from phase structure (EoS) existing and surviving inside the evolution’s final snapshot”, and if “yes”, then “how to exclusively and effectively decode these information from the complex final output”, but NOT as what the referee said in the above comment.

Secondly, the large number of options for architectures of deep convolution neural network entailed us to employ the most commonly used blocks that has been proven useful, such as the BatchNormalization, Dropout and L2 regularization. The current architecture could be improved, but for our purpose – the proof of principle, a generalization performance with 95% recognition accuracy is perfectly fine for our motivation in the study for demonstration.

Thirdly, we did try some simpler machine learning methods in the beginning but only got limited performance. We show that the event-by-event distribution of a few important observables – $\langle p_T \rangle$, v_n and event angles ψ_n – do overlap for 2 different EoS cases and can not be distinguished by the traditional methods developed in high-energy heavy-ion science, as shown in Fig. 1, 2 and 3 (in this reply).

The referee’s suggestions for a much thorough analysis using different machine learning techniques for the data structure is relevant for further works. We computed the correlation matrix of 85 observables/features which have been designed in the last two decades in the field. Here we provide for the first time

a systematic study using many machine learning techniques on the event-by-event data from relativistic heavy-ion collisions. The correlation matrix of pairs of important distinct observables is shown in Fig. 4 (in this reply),

	GROUP1	GROUP2
obs + Gaussian Naive Bayes	46.2%	47.6%
obs + Decision Tree	57.5%	64.9%
obs + Random Forest	62.5%	69.8%
obs + Gradient Boosting Trees	66.9%	81.9%
obs + linear SVC	75.8%	84.6%
obs + SVC rbf kernel	60.9%	56.7%
raw + linear SVC	65.2%	84.3%
pca + linear SVC	46.4%	47.7%

Table 1: Prediction accuracies from traditional machine learning algorithms using scikit-learn. Where “obs” stands for 85 pre-defined observables, “raw” stands for $\rho(p_T, \Phi)$ and “pca” denotes the first 150 components from principle components analysis (PCA) on raw spectra.

Of the traditional machine learning tools, we have systematically checked the prediction accuracies with various classifiers using scikit-learn. As shown in Table. 1 (in this reply), the best performance (with good generalizability) is obtained by a linear support vector machine classifier (linear-SVC) – on average 80% prediction accuracy using the best estimator from a grid search on pre-defined observables. The ensemble methods random forest and gradient boosting trees get higher score than a single decision tree. Gaussian Naive Bayes classifier and pca + linear SVC have poor generalizability on two groups of testing data – with less than 50% accuracy.

For the trained linear-SVC classifier with pre-defined observables, we list the 20 most important features in descending order – ‘ptspec-bin4’, ‘ptspec-bin5’, ‘ptspec-bin8’, ‘ptspec-bin7’, ‘ptspec-bin6’, ‘ptspec-bin1’, ‘ dN/dy ’, ‘ptspec-bin2’, ‘ptspec-bin3’, ‘ptspec-bin11’, ‘v2-ptbin5’, ‘v2-ptbin6’, ‘v2-ptbin4’, ‘ptspec-bin9’, ‘v5-ptbin12’, ‘ptspec-bin10’, ‘v5-ptbin11’, ‘ptspec-bin12’, ‘ptspec-bin0’, ‘v2-ptbin7’.

Apparently both the shape of the p_T spectra and the azimuthal angle distribution at different p_T s dominated by the bins 4 - 8 (correspond to p_T range 0.3 ~ 1.0 GeV) are decisive for the classification capability of this linear-SVC. Trained with the pre-defined observables, the shape of the p_T spectra is more important than the azimuthal angle distribution. The important p_T range obtained from linear-SVC agrees with the importance map from the prediction difference analysis of the deep convolution neural network.

The important features from linear-SVC trained with raw spectra is shown in Fig. 5 (in this reply). The p_T range in this importance map covers part of that from deep CNN. However, the most important features along the azimuthal angle direction center at $\Phi = \pi/2$ and $\Phi = 3\pi/2$. As we want to confirm that the dynamical evolution is indeed encoded in the final state particle spectra, a deep neural network works much better as the prediction accuracy is much

higher and the importance region follows intuition.

We do agree with the referee that not only deep learning, but also some traditional big data analysis with machine learning tools now bring revolution to the heavy-ion community, as relativistic heavy-ion collisions have accumulated a huge amount of data. Here we have demonstrated the power of both, machine learning and deep learning, for high energy physics. The whole analysis script in jupyter notebook will be open sourced at the time of publication of this article.

We have added these plots to the supplementary materials.

Referee’s comment: *Indeed, I find it highly suspect to use a convolutional model in this case. Is it really the case that there is translation invariance in both axes? The response to reviewers claim there is invariance to Φ , but Figure 2 seems to contradict this. At a basic level, why is convolution even necessary? It provides a more compact parameterization, but the “images” here are tiny and an arbitrary amount of data is available, so why not train a fully-connected network and evaluate how it compares? This is just one example of how such a paper needs to peel back the layers of modeling complexity to try to identify what is providing information about the classification. How can we feel comfortable that the classification is not simply arising from some very simple first or second order statistics in the data? Deep learning methods are only interesting when other simpler things fail.*

Authors’ reply: There is indeed translation invariance in the features that we use to identify the EoS. Our input “image” is the hadron density in azimuthal angle and transverse momentum p_T . The features are hadron distributions relative to the event-plane angles Ψ_n (where Ψ_2 is the orientation angle of the elliptic mode and Ψ_3 is the orientation angle of the triangular mode). These features in the relative distributions are independent of the values of Ψ_n . We did indeed also try fully connected neural networks in the beginning when we did not have enough data (even with GPU parallelization, it takes much longer to accumulate enough training data than the training process), we noticed that CNN has much better performance than fully connected neural network. Even with 22000 events, the best performance of FC networks, with 2 to 5 hidden layers, gave on average 90% recognition rate on the testing data. Data augmentation in FC networks bring negligible improvement (less than 1%) on the testing data. FC networks neglect the translation invariance of the local correlations of particles that are close to each other in momentum space. Hence, CNN is not only compact, but also works better due to the translational invariance in momentum space. The translational invariance can also be seen by comparing to the important features as shown in Fig. 5 (in this reply) from a poor linear-SVC classifier.

The following statement is appended to the method section to discuss the translational invariance in the momentum-space image,

“The input images in the present method are the particle density distributions in momentum space. Due to collective expansion of the QGP, fluctuations at initial state transform to strong correlations of final state particles in the images. These local structures and translational invariance of odd-order Fourier

decomposition along the azimuthal angle direction make convolution neural networks preferable to fully connected neural networks.”

Referee’s comment: *To summarize, I feel that much more empirical rigor is needed here if this work is intended to inspire further experimental studies. In my opinion it is not sufficient to simply show that there exists a complicated function approximator to this problem; rather, one must tease apart the characteristics of failed and successful such approximators to reveal insights into the underlying phenomena.*

Authors’ reply: Teasing apart characteristics of failed and successful features is a good suggestion. We are very interested in understanding the physics explanations behind the deep learning results. The prediction difference analysis does, indeed, tell us which features are most important for classification. Results from a support vector machine also support the finding of convolution neural networks, namely that the most important features are the angular distributions of particles at low and intermediate transverse momenta. However, in pre-defined observables, the distribution in Φ , for these p_T bins are either integrated out or Fourier transformed. It is found that traditional machine learning algorithms, which work on these pre-defined observables, can NOT achieve an accuracy given by the CNN.

We hope that with the additional information given and the provided figures on conventional observables, we have convinced the referee that applying deep learning methods to event-by-event raw heavy ion data constitutes an important step forward to obtain insights on important questions, like identifying the order of the QCD phase transition.

A list of changes to the manuscript

In response to the first referee’s question, we have made the following changes,

Change 1) In TABLE II, we have added more events for TESTING DATASET GROUP 2 with $\eta/s = 0.08$.

Change 2) In TABLE III, we have updated the testing accuracy and model uncertainty for GROUP 2.

Change 3) In FIG. 2, we have added importance maps (g) and (h) for GROUP 2 with $\eta/s = 0.08$.

In response to the second referee’s questions, we have made the following changes,

Change 4) We have added the following discussions in the beginning of Section III,

“Besides the deep CNN method employed in the present paper, there are also some other machine learning classifiers. In the supplementary materials we

attached the results from several traditional machine learning methods, such as support vector machine classifier (SVC), decision trees, random forests and gradient boosting trees. The best classifier (linear SVC) that generalizes well on two testing datasets achieves on average $\sim 80\%$ prediction accuracy. The important features from different classifiers differ from each other, however, those with good generalization capability have similar importance regions as given by the deep CNN. The deep CNN with on average $\sim 95\%$ prediction accuracy works much better to answer the core questions – “Is there a traceable encoder of the dynamical information from phase structure (EoS) that survives the evolution and exists in the final snapshot?” and if “yes”, then “how to exclusively and effectively decode these information from the highly complex final output”. These questions are crucial but unclear for decades in high-energy heavy-ion physics (and also in physical cosmology) due to the complexity and highly-dynamical characteristics in the collision evolution. The deep CNN demonstrates the revolution that big data analysis and machine learning might bring to the high energy physics and astrophysics.”

Change 5) We have added the following discussions in the end of Section IV,

“The input images in the present method are the particle density distributions in momentum space. Due to collective expansion of the QGP, fluctuations at initial state transform to strong correlations of final state particles in the images. These local structures and translational invariance of odd-order Fourier decomposition along the azimuthal angle direction make convolution neural networks preferable to fully connected neural networks.”

Change 6) We have uploaded the training and testing data to Figshare and have added the following “Data Availability” in the method section.

“The datasets generated and analysed during the current study are available in the public repository [48], <https://doi.org/10.6084/m9.figshare.5457220.v1>.”

Change 7) We have added the online repository that hosts the data used in the present paper,

[48] L. G. Pang, K. Zhou, N. Su, P. Hannah, H. Stocker, X. N. Wang, Training and testing data used in the paper “An equation-of-state-meter of QCD transition from deep learning”, figshare, <https://doi.org/10.6084/m9.figshare.5457220.v1>

Change 8) We have added a new section in the supplementary material to demonstrate the performance of big data analysis using traditional machine learning methods.

Figure 1: The event-to-event distribution of mean p_T and v_n which captures the particle distribution along radial and azimuthal angle direction.

Figure 2: The event-to-event distribution/fluctuation of the event planes Ψ_n .

Figure 3: The scatter plots between several pairs of observables.

Figure 4: The correlation matrix between $\langle p_T \rangle$, v_2 , v_3 , v_4 , v_5 and dN/dY on testing data GROUP1, reveals various correlations that were found one after another in the last two decades.

Figure 5: The important features from a linear-SVC trained with raw spectra $\rho(p_T, \Phi)$.

REVIEWERS' COMMENTS:

Reviewer #1 (Remarks to the Author):

The authors have improved their manuscript by taking into account the optional suggestion I made. As I have stated in my previous comment, the idea presented in the manuscript is important enough from the viewpoint of heavy-ion physics and thus I recommend it for publication in Nature Communications.

Reviewer #5 (Remarks to the Author):

There are three main goals of the paper contained in the text: G1) (page 3) "We find unique encoders of bulk properties inside particle spectra in terms of high-level representations using deep-learning techniques, which are not captured by conventional observables"; G2) (page 3) "The performance is surprisingly robust against ..." some set of simulation parameters.; G3) (page 4) "The network we develop below is, however, not sensitive to the model-dependent features." It is quite informative results which give a good example of the DL importance for the community of physicists. Roughly speaking, paper clearly demonstrate that DL with the simulation data obtained with one algorithm can be used for the successful analysis of the simulation data obtained by some other algorithm. It is a hope that CNN trained on the simulation data will be able to analyze experimental data. This hope is based on the assumption that simulation model really capture physics of the experiment. In my opinion, it is the only logical gap in the paper.

I found paper is interesting for the physicists. Of course, it is a beginning of the analysis, and questions which arises in the discussion with referees have to be answered in the future research.

I recommend paper for publications. It contains new important information on the possible classification of phase transitions in QCD based on the classification of simulation and experimental data using DL/CNN approach.

Reviewer #6 (Remarks to the Author):

I have read the last version of your submission, as well as the last comments of reviewer#2 and your replies. Substantially, I agree with your point of view: Checking that a CNN can discriminate between the states corresponding to the $\rho(p_T, \Phi)$ "images" is interesting enough to merit publication, given that your CNN design is acceptable and that shallow classification machines and ensembles offer worse performance.

However, let me recommend:

- 1.- It is more than appropriate that you include in your manuscript the mention to the experimental work you carried out with fully connected schemes, getting worse results when you had less examples.
- 2.- I recommend to explain –at least qualitatively– how you established the different non-trainable parameters of your machine design, from the size of the sub-images to the number of CNN layers, size of the final (pre-output) layer, drop-out rate, etc. In a future, to determine the performance sensitivity with respect to these parameters will be necessary.

And three suggestions for further work:

- 1.- Data augmentation has proved to be of moderate benefit with some fully connected machines. Is this also true for CNNs? Note that there are many "ad hoc" data augmentation techniques.
- 2.- In my experience, data augmentation is effective when working with Deep Belief Nets and Stacked Denoising Auto-Encoding classifiers: Even simple noise learning helps. Since Auto-encoding can be relevant in order to understand how classification is done, what about to explore this alternative way of dealing with your problem?
- 3.- In any case, pre-emphasis techniques (similar to those applied for boosting, but applied in one shot after a preliminary classification) usually improve deep classifier performances. I suggest you to explore this avenue, too.

Reviewer 5

Referee's comment: *(Remarks to the Author): There are three main goals of the paper contained in the text: G1) (page 3) We find unique encoders of bulk properties inside particle spectra in terms of high-level representations using deep-learning techniques, which are not captured by conventional observables; G2) (page 3) The performance is surprisingly robust against . . . some set of simulation parameters.; G3) (page 4) The network we develop below is, however, not sensitive to the model-dependent features. It is quite informative results which give a good example of the DL importance for the community of physicists. Roughly speaking, paper clearly demonstrate that DL with the simulation data obtained with one algorithm can be used for the successful analysis of the simulation data obtained by some other algorithm. It is a hope that CNN trained on the simulation data will be able to analyze experimental data. This hope is based on the assumption that simulation model really capture physics of the experiment. In my opinion, it is the only logical gap in the paper. I found paper is interesting for the physicists. Of course, it is a beginning of the analysis, and questions which arises in the discussion with referees have to be answered in the future research. I recommend paper for publications. It contains new important information on the possible classification of phase transitions in QCD based on the classification of simulation and experimental data using DL/CNN approach.*

Authors' reply: We thank the referee for positive comments and recommendation for publication. We totally agree with the referee that applying DL models trained with simulated data to true experimental data is a further step to go. The situation is quite similar to training robots in a virtual reality environment that implements all known physical laws. On one hand, the relativistic fluid-dynamics is proven to be the most successful model to describe the evolution of the Quark Gluon Plasma and many collective observables of produced hadrons in heavy ion collisions. On the other hand, we know that there are causality links between the equation of state, the pressure gradient and the momentum distribution of final state hadrons. Supervised learning using deep neural network will assist physicists to identify these links more easily. The biggest obstacle might be the efficiency problem where experimental detectors can only capture $\sim 80\%$ final state hadrons, this will be fixed in the future by adding detector simulations in the model. The optimal situation would be that the signal is strong enough such that randomly dropping 20% of the particles do not affect model performance. This is rather true for our brain and the convolution neural network since they are quite robust to the resolution of the images.

Reviewer 6

Referee's comment: *I have read the last version of your submission, as well as the last comments of reviewer#2 and your replies. Substantially, I agree with*

your point of view: Checking that a CNN can discriminate between the states corresponding to the $\rho(p_T, \Phi)$ images is interesting enough to merit publication, given that your CNN design is acceptable and that shallow classification machines and ensembles offer worse performance. However, let me recommend: 1.-It is more than appropriate that you include in your manuscript the mention to the experimental work you carried out with fully connected schemes, getting worse results when you had less examples.

Authors' reply: We thank the referee for the positive comments and recommendation for publication. We have included the following content in the manuscript:

The relativistic hydrodynamic simulations of the heavy ion collisions are quite computing intensive, even with the GPU parallelization, it still takes much longer to accumulate enough training data than the training process. In the beginning of this study when the training data size is not big enough, we experimented with fully connected neural networks. However, the network always overfits the training data and fails to work with the validating data. We noticed that CNN has much better generalizability than fully connected neural network with small set of data. With 22000 events, the best performance of fully connected neural networks, with 2 to 5 hidden layers, gave on average 90% recognition rate on the testing data. Data augmentation in fully connected neural networks bring negligible improvement (less than 1%) on the testing data. The fully connected neural networks neglect the translation invariance of the local correlations of particles that are close to each other in momentum space.

Referee's comment: *2.-I recommend to explain at least qualitatively how you established the different non-trainable parameters of your machine design, from the size of the sub-images to the number of CNN layers, size of the final (pre-output) layer, drop-out rate, etc. In a future, to determine the performance sensitivity with respect to these parameters will be necessary.*

Authors' reply: Thanks for this suggestion, the referee is right that explaining how we fix those non-trainable parameters may help future applications to scientific problems. We have added the following in the manuscript:

There are several non-trainable parameters in the neural network, such as the number of hidden layers, the size of the convolution kernels, the size of the final hidden layer and the dropout rate. The neural network in the present work can be easily rebuilt with these hyper-parameters in Keras (the source code is also available as requested). These parameters are adjusted heuristically to maximize the training accuracy and validation accuracy but not the testing accuracy. The first step is to choose the number of hidden layers, the size of the convolution kernels and the size of the final hidden layer such that the model has enough capacity to describe the training data. At this step, we use a small portion of the training data, tune the widely used values of parameters and observe big training accuracy but small validation accuracy. It is found that the widely used convolution kernel sizes 5×5 and 3×3 do not work well at this step and increasing the number of the convolution layers from 2 to 3 does

not improve the training accuracy and the validation accuracy. The next step is to increase the validation accuracy, in addition to the batch normalization and L2 regularization, it is found that dropout with a proper rate and tuning the size of the final hidden layer help to increase the validation accuracy. With this minimal working neural network, the validation accuracy increases rapidly with more training data. What is interesting is that when there are big training data, the previously not functioning architectures (with smaller convolution kernels and more hidden layers) also start to work and produces similar testing accuracy. The optimal neural network architecture and the values of the non-trainable parameters with big training data may desire future investigation.

Referee's comment: *And three suggestions for further work: 1.-Data augmentation has proved to be of moderate benefit with some fully connected machines. Is this also true for CNNs? Note that there are many ad hoc data augmentation techniques. 2.-In my experience, data augmentation is effective when working with Deep Belief Nets and Stacked Denoising Auto-Encoding classifiers: Even simple noise learning helps. Since Auto-encoding can be relevant in order to understand how classification is done, what about to explore this alternative way of dealing with your problem? 3.-In any case, pre-emphasis techniques (similar to those applied for boosting, but applied in one shot after a preliminary classification) usually improve deep classifier performances. I suggest you to explore this avenue, too.*

Authors' reply: Thanks the referee for the suggestions on future work. We agree with the referee that there are a lot of interesting avenues that can be explored to improve the performance of the classifier. During the research, we have noticed that the good generalizability of deep convolution neural network may be connected to the symmetry restoration picture in physics. The convolution operation restores the translational invariance, the pooling restores scaling invariance, multiple convolution kernels (multiple matrix multiplications) restores the rotational invariance. Various data augmentation techniques together with the noise help to restore more irrelevant broken symmetries. The more symmetries are restored, the better generalizability the deep CNN may have. We wish to explore this direction in the future. Layer-by-layer unsupervised pre-training using Stacked Denoising Autoencoder may help to preserve the most relevant broken symmetries(features) during the re-construction step at each layer. This method together with the generative adversarial network will help us a lot in semi-supervised learning with insufficient labeled data.